# Development of a minigenome cassette for Lettuce necrotic yellows virus: A first step in rescuing a plant cytorhabdovirus

**Ahmad E. C. Ibrahim, Craig J. van Dolleweerd, Pascal M. W. Drake**[ID]◉*, **Julian K-C. Ma**◉

Institute for Infection and Immunity, St. George's University of London, London, United Kingdom

◉ These authors contributed equally to this work.
* pdrake@sgul.ac.uk

## Abstract

Rhabdoviruses are enveloped negative-sense RNA viruses that have numerous biotechnological applications. However, recovering plant rhabdoviruses from cDNA remains difficult due to technical difficulties such as the need for concurrent *in planta* expression of the viral genome together with the viral nucleoprotein (N), phosphoprotein (P) and RNA-dependent RNA polymerase (L) and viral genome instability in *E. coli*. Here, we developed a negative-sense minigenome cassette for Lettuce necrotic yellows virus (LNYV). We introduced introns into the unstable viral ORF and employed *Agrobacterium tumefaciens* to co-infiltrate *Nicotiana* with the genes for the N, P, and L proteins together with the minigenome cassette. The minigenome cassette included the *Discosoma* sp. red fluorescent protein gene (DsRed) cloned in the negative-sense between the viral *trailer* and *leader* sequences which were placed between hammerhead and hepatitis delta ribozymes. *In planta* DsRed expression was demonstrated by western blotting while the appropriate splicing of introduced introns was confirmed by sequencing of RT-PCR product.

## Introduction

Lettuce necrotic yellows virus (LNYV) is a plant cytorhabdovirus with a negative-sense, single-stranded, RNA genome that infects lettuce and garlic [1, 2]. *In planta*, the virus replicates within the cytoplasm accumulating in membrane-bound viroplasms [3, 4]. LNYV genomic RNA (gRNA) encodes six monocistronic genes: Nucleoprotein (N), Phosphoprotein (P), cell-to-cell movement protein (4b), Matrix protein (M), Glycoprotein (G), and Large Polymerase (L), flanked by the untranslated 3' *leader* and 5' *trailer* regions [5]. As with other members of the *Rhabdoviridae* family, gRNA coiled with N protein acts as the template for *de novo* mRNA synthesis by the viral RNA-dependent RNA polymerase [6]. The viral transcription and replication processes are controlled by *cis*-acting sequences in the *leader* (*le*) and *trailer* (*tr*) regions [7–9].

The inherent characteristics of rhabdoviruses, such as having well-defined transcription start/stop signals [10] and the ability to incorporate recombinant glycoprotein into their

**Data Availability Statement:** All relevant data are within the paper and its Supporting Information files.

**Funding:** This work was supported by a Wellcome Trust grant (JM) (Grant number WT093092MA, https://wellcome.ac.uk), the Hotung Charitable Settlement (JM) and an ERC award (JM) (Grant number ERC-2010-ADG_20100317, https://erc. europa.eu/). The funders had no role in study design, data collection and analysis, decision to publish, or preparation of the manuscript.

**Competing interests:** The authors have declared that no competing interests exist.

envelope [11], make them attractive candidates for various biomedical applications. For example, established reverse genetic systems for animal rhabdoviruses [12, 13] enabled rescue of vesicular stomatitis virus (VSV) as recombinant viral vaccine [14, 15]. In such systems, cells are transfected with plasmids encoding the viral genome together with the N, P, and L proteins (helper proteins) [16]. The viral genome is provided in the positive (antigenomic) sense [17] and auto-cleaving ribozymes such as hammerhead [18] and hepatitis delta [19] are used to increase yield by generating the exact termini of the *de novo* transcribed viral RNA. Recently, this system was adapted for the plant nucleorhabdovirus, Sonchus yellow net rhabdovirus (SYNV), where *Agrobacterium tumefaciens* was used to deliver plasmids encoding the antigenomic-sense of the viral genome together with the N, P, and L proteins [20].

Here we attempted to adopt such a system to rescue LNYV. We employed a minigenome (MG) system [21, 22], in which the full-length viral cDNA was substituted by that of a reporter gene cloned between the viral le and tr sequences. Minigenome-based systems are a rapid and simple approach for proof-of-concept studies. As with full-virus RNA transcripts, the MG-derived reporter gene RNA transcripts undergo transcription and replication by the helper proteins. Detection of the reporter protein demonstrates the productive interaction between the viral *cis*- and *trans*- acting elements. We assembled LNYV negative-sense MG cassette with *Discosoma* sp. red fluorescent protein gene (DsRed) cloned in the negative-sense between LNYV le and tr sequences and hammerhead (HHz) and hepatitis delta virus (HDz) ribozymes at the 5' and 3' termini respectively, in the plant expression vector pTRAk.2 [23]. Similarly, we cloned the N and P genes open reading frame(s) (ORF) into pTRAk.2.

However, our initial attempts to clone the L gene ORF into pTRAk.2 were hampered due to its instability in *E. coli*. Viral sequence instability in *E. coli* has been observed previously where bacteria harbouring plasmids with viral sequences fail to grow or yield plasmids with nucleotide mutations. The reason for this instability remains unknown but different strategies such as cell-free cloning [24] or intron introduction [25, 26] have been employed to circumvent this problem. In the present study, we overcame L gene instability by introducing three introns and using an *E. coli* strain that enabled downregulation of plasmid copy-number.

Following successful molecular cloning of all four constructs, we used *A. tumefaciens* to co-deliver the plasmids into *Nicotiana glutinosa* and *Nicotiana benthamiana*. The former is known to be susceptible to LNYV infection and has been the plant of choice for studying this virus biology and pathogenesis while the latter is the species of choice for production of recombinant proteins in plants using *Agrobacterium*-mediated transient expression [27, 28].

## Materials and methods

### Bacterial strains

*E. coli* strain MAX Efficiency® DH10B™ (Thermo Fisher Scientific, UK) and *E. coli* strain CopyCutter™ EPI400™ (Epicentre, US) were used for molecular cloning. For plant infiltrations, *A. tumefaciens* strain GV3101::pMP90RK (German Collection of Microorganisms and Cell Cultures (DSMZ), Leibniz Institute, Germany) was employed.

### Molecular biology

Plasmid DNA extraction was undertaken using the QIAprep Spin Miniprep (Qiagen, Manchester, UK). DNA restriction endonucleases and ligase were purchased from New England Biolabs (NEB) (Hitchin, UK). Chemicals were purchased from Sigma Aldrich (Dorset, UK) or Thermo Fisher Scientific (Dartford, UK). PCR amplifications were completed using Phusion® High-Fidelity PCR Master Mix with HF Buffer or with Q5 Master Mix (NEB).

## Gene constructs

LNYV cDNA was synthesized by GeneArt Gene Synthesis (Regensburg, Germany) as separate constructs with sequences cloned in the opposite sense to the antibiotic resistance gene in the plasmid. Constructs for MG cassettes, N, and P genes were generated using PCR with appropriate DNA oligonucleotides and cloned into the pTRAk.2 plasmid using compatible restriction endonuclease sites. Similarly, the L gene with three introduced introns was synthesized by Genscript (New Jersey, USA) and cloned into the pTRAk.2 plasmid. The pTRAk.2 plant transformation vector includes (i) Cauliflower mosaic virus (CaMV) promoter with duplicated transcriptional enhancer and (ii) CaMV-35S transcription polyadenylation signals for *in planta* expression [23]. The gene for DsRed protein with transit peptide sequence at its N—terminus was PCR-amplified from pTRAp-2G12-Ds plasmid with appropriate DNA oligonucleotides and cloned into assembled *in vitro* MG LNYV construct using compatible restriction endonuclease sites. The DsRed gene used was that of *Discosoma* sp. red fluorescent protein (R2G mutant) ORF with *H. vulgare* granule-bound starch synthase I transit peptide sequence at its N–terminus.

## In vitro transcription

For *in vitro* assessments, a DNA fragment containing the minigenome cassette was amplified by PCR using priV set of primers (S1 Table) from an assembled construct covering (i) the T7 promoter, (ii) the hammerhead ribozyme, (iii) the LNYV tr, (iv) the multiple cloning site, (v) the LNYV le, (vi) hepatitis delta ribozyme, and (vii) T7 terminator. The PCR amplicon was used as a template for *in vitro* transcription using the TranscriptAid T7 High Yield Transcription Kit (Thermo Fischer Scientific) which was incubated at 25°C in the presence of 11 mM $MgCl_2$ for two hours to allow ribozyme activity as described previously [22, 29].

## Plant agroinfiltration

*N. glutinosa* and *N. benthamiana* plants were grown at 23°C ± 2°C in a 16/8 h light and dark cycle and agro-infiltrated using *A. tumefaciens* strain GV3101::pMP90RK as described previously [30].

## Total RNA extraction and RT-PCR analysis

Total RNA was extracted from leaf tissue 7 days post infiltration using TRI Reagent® Solution (Applied Biosystems, UK). Extracted RNA was treated with r*DNase*I using Ambion® DNA-*free*™ Kit (Thermo Fischer Scientific). RT-PCR amplifications were performed using Access RT-PCR System from Promega (Southampton, UK) according to the manufacturer's protocol. RT-PCR transcribed cDNA was sequenced (Genewiz, Bishop's Stortford, UK).

## Western blot analysis

Infiltrated and non-infiltrated plant leaf samples (100 mg) were frozen in liquid nitrogen, homogenized on ice with plastic pestles in PBS pH 7.4 buffer. Crude protein samples were heated in NuPAGE® LDS Sample Buffer with 500 mM dithiothreitol (DTT) (Thermo Fischer Scientific) at 70°C for 10 minutes and loaded onto an Invitrogen™ NuPAGE® Novex® Bis-Tris 4–12% gel (Thermo Fischer Scientific).

Separated proteins were then blotted onto Amersham Hybond ECL Nitrocellulose Membrane (GE Healthcare, Amersham, UK) using Invitrogen™ semi-dry transfer system (Thermo Fischer Scientific). To detect DsRed protein, the membrane was probed with mouse monoclonal [6G6] anti-red fluorescent proteins (ChromoTek, GmbH, Planegg-Martinsried, Germany).

Antibody 6G6 has been used previously to detect red fluorescent protein in different plant expression platforms such as *Nicotiana benthamiana* [31] and *Arabidopsis thaliana* [32].

Antibody 6G6 was diluted at 1:2500 in 5% w/v non-fat dried skimmed milk (NFDS) in PBST (0.1% v/v Tween 20 PBS pH 7.4 buffer) and incubated with the nitrocellulose membrane overnight at 4˚C. This was followed by sheep anti-mouse IgG-HRP conjugate (The Binding Site, UK) diluted 1:5000 in 5% w/v NFDS milk in PBST and incubated for 1 hour at room temperature. The membrane was washed three times with in PBST after each step. The membrane was developed by Amersham™ ECL™ Prime Western blotting reagent (GE Healthcare) and visualised using SynGene G:Box gel imaging technology with Genesys software (Syngene, UK).

## Results

### Introns stabilise L gene ORF in E. coli and are correctly spliced in planta

We previously attempted conventional methods to clone the L gene ORF in *E. coli* but we could not recover the correct construct. Based on the observation that introns stabilise viral genomes in *E. coli* [25] we introduced the second intron of the *Phaseolus vulgari* nitrite reductase (NIR) gene [33], the second intron of the *Solanum tuberosum* light-inducible tissue-specific ST-LS1 gene [34], and finally the first intron of *Nicotiana tabacum* ribulose 1,5-bisphosphate carboxylase small subunit (NtRbcS) gene [35] into the L gene ORF (Fig 1A). Furthermore, the *E. coli* strain CopyCutter™ EPI400™ was employed to maintain low construct copy numbers. The L gene (with introns) was successfully cloned into the plant expression plasmid pTRAk.2 (int-L.pTRAk.2) using this strategy (confirmed by Sanger sequencing).

The fidelity of intron splicing was subsequently tested *in planta*. Total RNA was extracted from *N. glutinosa* infiltrated with int-L.pTRAk.2 and used as a template for RT-PCR using prin1, prin2, and prin3 set of primers (S1 Table). The primers were designed to amplify three segments spanning introns I, II, and III respectively. The expected cDNA size of each segment with introduced introns was 500 bp, while 279, 312, and 407 bp respectively if introns were spliced out (Fig 1A). As shown in Fig 1B, all three introns were removed giving bands of the expected size. Furthermore, the correct splice-site sequence was confirmed by Sanger sequencing.

### A minigenome system for LNYV

Initially, we assembled LNYV *in vitro* MG cassettes to assess ribozyme auto-cleavage in an *in vitro* transcription assay. The MG cassette consisted of tr–multiple cloning site (MCS)–le cloned between hammerhead and hepatitis delta and under the control of T7 promoter (Fig 2A). We also changed the first 8 nucleotides of HHz (accession number GU299211) to 5'—UCGUCCGU–3', to allow base-pairing with the first 8 nucleotides of the LNYV tr (5'—ACGG ACGA–3') as indicated in a previous study [18].

As shown in (Fig 2B), gel electrophoresis of the transcribed products yielded a fragment of expected size for the tr–MCS–le sequence (358nt) suggesting auto-cleavage by both HHz and HDz ribozymes. Putative bands comprising HHz–tr–MCS–le (412nt) and HDz–T7 (233nt) terminator were also seen.

We next sought to test the LNYV MG cassette *in planta* to assess the interaction between LNYV *cis-* and *trans-* acting elements. We substituted the MG MCS with the DsRed gene ORF, cloned in the negative-sense with a plastid targeting sequence at the N–terminus. The plastid targeting sequence was employed to minimize metabolic burden on the endoplasmic reticulum [36]. The coding region was cloned between LNYV L gene 3' UTR (extending between 12,564 and 12,613nt of LNYV genome accession number AJ867584) and N gene 5'

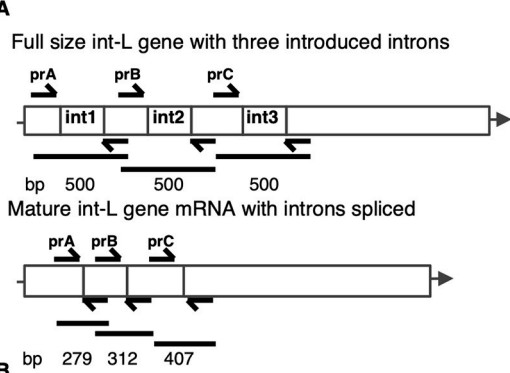

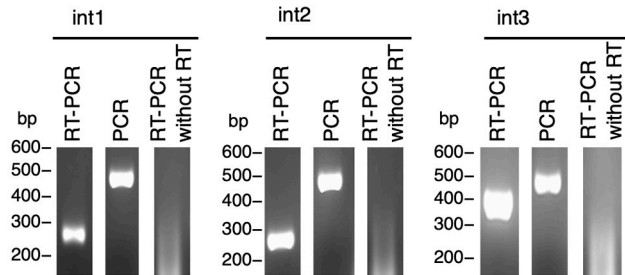

**Fig 1. Genome organization of LNYV L gene with or without introduced introns.** (**A**) Diagram showing the expected size for RT-PCR amplicons from int-L gene pre- and mature mRNA using prA, prB, and prC primers. (**B**) Agarose gels int1, int2, and int3 showing mRNA amplicons detected by RT-PCR compared to PCR of the int-L gene for second intron of the *Phaseolus vulgari* nitrite reductase (NIR) gene, the second intron of the *Solanum tuberosum* light-inducible tissue-specific ST-LS1 gene, and the first intron of *Nicotiana tabacum* ribulose 1, 5-bisphosphate carboxylase small subunit (NtRbcS) gene respectively. A RT-PCR without addition of reverse transcriptase enzyme was used as a negative control.

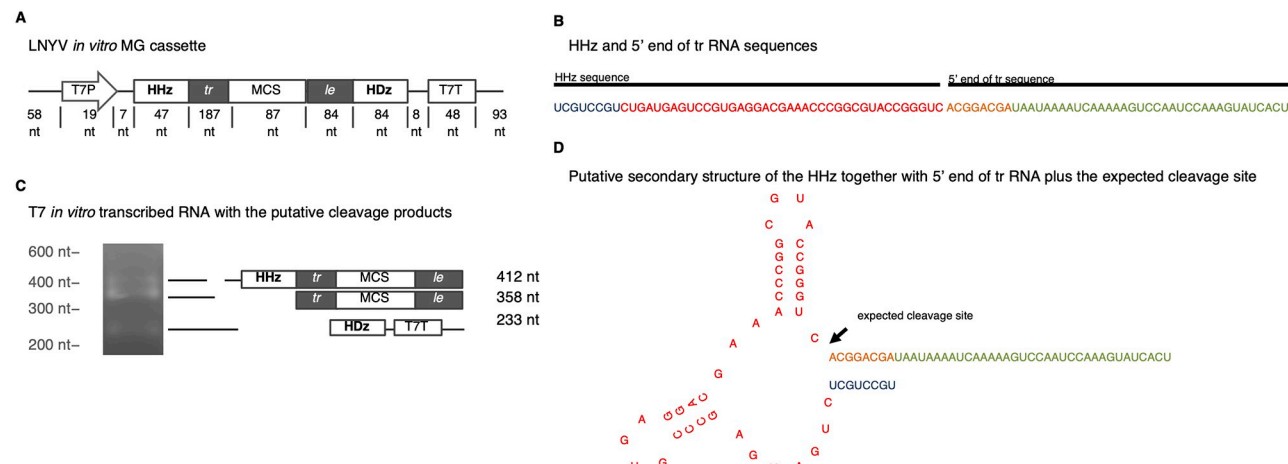

**Fig 2. Testing ribozyme auto-cleavage in the LNYV minigenome cassette.** (**A**) Diagram of construct used in T7 *in vitro* RNA transcription showing the LNYV minigenome cassette tr–MSC—le flanked by hammerhead and hepatitis delta ribozymes for generating exact viral termini by auto-cleavage. The minigenome cassette is flanked by the T7 promoter and terminator for *in vitro* transcription. (**B**) RNA sequence of HHz and the 5' end of tr used in the minigenome cassette. The first 8 nucleotides of HHz were changed to allow base-pairing with the first 8 nucleotides of the tr sequence. (**C**) Agarose gel showing putative bands of tr–MCS—le fragment (358 nt), HHz—tr–MCS—le (412 nt), and HDz—T7T (233 nt). (**D**) The expected secondary structure of the HHz with its first 8 nucleotides complimenting those of the tr sequence plus the putative cleavage site [18].

UTRs (extending between 92 and 168nt). The construct was cloned into the pTRAk.2 plant expression plasmid (Fig 3A) under the control of the cauliflower mosaic virus 35S double promoter for transient expression in *Nicotiana* [37]. N, P and int-L genes were cloned in separate plasmids.

*N. glutinosa* leaves were co-infiltrated with the MG cassette, N, P and int-L. 7 days post-infiltration, total RNA was extracted from co-infiltrated leaf samples and used as a template for RT-PCR using N, P, int-L, and DsRed gene-specific primers (sets prRN, prRP, prRint-L and prRR respectively). Transcription of DsRed, N, P and L genes was confirmed by detection of bands of expected size (Fig 3B). Two bands were detected for the int-L gene mRNA (lane int-L). The smaller-sized band is consistent with int-L mRNA having all introns spliced out (expected size 1,955 bp) whilst the larger band is consistent with the int-L gene pre-mRNA containing introns I and II (expected size 2,365 bp).

Plant samples were subsequently assayed for DsRed expression. Chlorophyll autofluorescence complicated detection of DsRed fluorescence, so western blotting with specific antiserum was used to confirm the expression of DsRed protein (33.5 kDa, Fig 4A). This was observed in *N. glutinosa* infiltrated with MG cassette, N, P and int-L plasmids (lane MG+-NPint-L) only. As shown in Fig 4A, no bands of DsRed protein could be seen in *N. glutinosa*

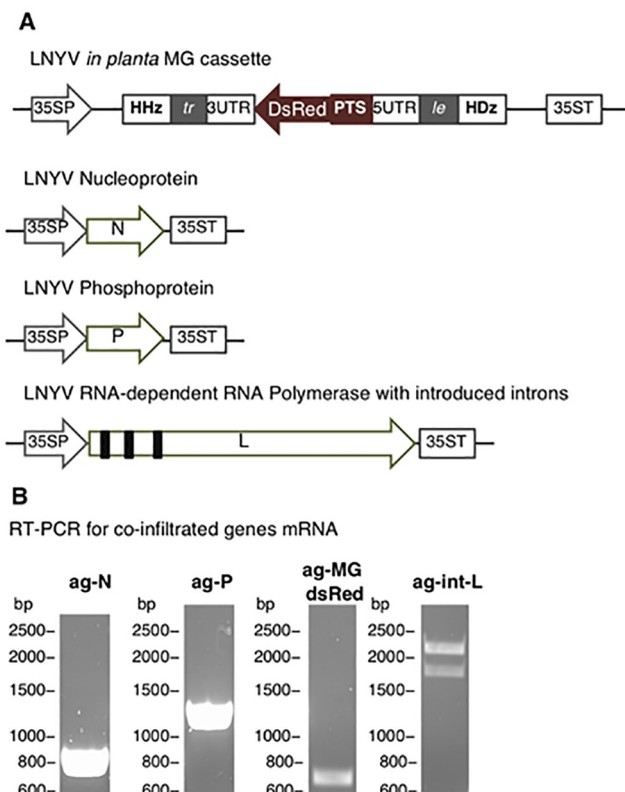

**Fig 3. Genome organization of LNYV minigenome cassette together with the N, P, and int-L gene used to develop reverse genetics for LNYV in *Nicotiana*.** (**A**) Diagram of LNYV minigenome construct used *in planta* together with the N, P, and int-L genes. All genes were flanked by cauliflower mosaic virus 35S double promoter and transcription polyadenylation signal (35SP and 35ST respectively) for in planta expression. The minigenome cassette constituted of (tr): LNYV trailer, 3'UTR: LNYV L gene 3' untranslated region, DsRed: DsRed gene ORF with plastid targeting signal (PTS) at N-terminus, 5'UTR: LNYV N gene 5' untranslated region cloned between hammerhead and hepatitis delta ribozymes (HHz and HDz respectively). (**B**) Agarose gels ag-N, ag-P, ag-MGdsRed, and ag-int-L showing the expected RT-PCR bands for N gene (889 bp), P gene (1,319 bp), DsRed (761 bp), and int-L genes (1,955 bp) respectively.

## A

Western blot on *N. glutinosa* infiltrated leaf sample

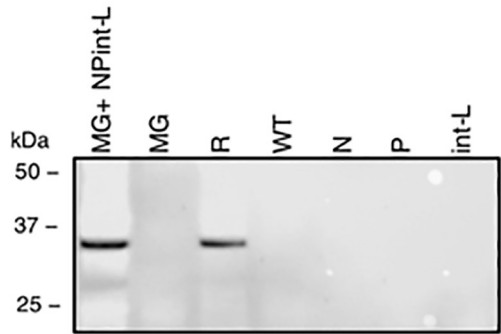

## B

Western blot on *N. benthamiana* infiltrated leaf sample

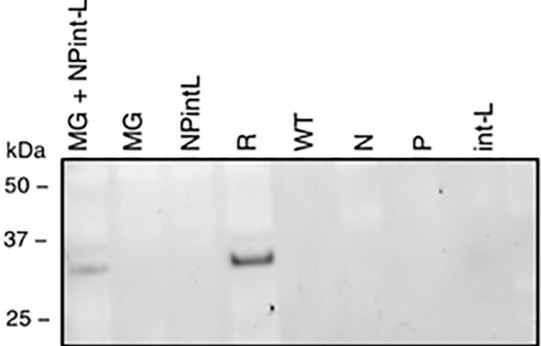

**Fig 4. Detection of DsRed in agroinfiltrated *Nicotiana*.** (**A**) DsRed protein detection using western blot in agro-infiltrated *N. glutinosa*. Infiltration was with MG cassette plus helper proteins (MG+NPint-L), MG alone (MG), helper proteins alone (N), (P), (int-L) and the positive control (R). WT is wild type non-infiltrated *N. glutinosa*. Bands were detected in samples infiltrated with MG cassette plus helper proteins (MG+NPint-L) and in positive control (R). (**B**) DsRed protein detection using western blot in agro-infiltrated *N. benthamiana*, samples are as in (**A**) but with an additional negative control (NPintL) consisting of infiltration of all helper proteins.

sample infiltrated with the MG cassette without helper plasmids (lane MG), samples infiltrated with individual helper plasmids without the MG cassette (lanes N, P, and int-L respectively) or in non-infiltrated *N. glutinosa* sample (lane WT). Furthermore, DsRed protein band was observed in *N. glutinosa* infiltrated with positive control construct (harbouring DsRed gene cloned in the positive-sense downstream CaMV35S promoter) (lane R) equivalent to that seen in lane (MG+NPint-L).

Similarly, western blotting of crude *N. benthamiana* extract samples confirmed the presence of DsRed protein in samples infiltrated with MG cassette, N, P and int-L plasmids (Fig 4B, lane mrR+NPint-L) and in the positive control (lane R). No bands were seen in *N. benthamiana* infiltrated with MG cassette only (lane MG), or in samples infiltrated with individual helper plasmids (lanes MG, N, P, and int-L respectively) or in non-infiltrated sample (lane WT). A

further negative control consisting of sample infiltrated with all three helper plasmids but without MG was included here and did not produce DsRed expression (lanes NPint-L).

## Discussion

In the present study, we have investigated the feasibility of a reverse genetic system for LNYV based on a negative-sense MG cassette. The MG-derived DsRed RNA-transcripts underwent transcription and expression in plant cells co-expressing N, P, and L proteins. Reverse genetic systems based on positive-sense constructs have been long established for animal and plant rhabdoviruses [12, 13, 22]. However, to the best of our knowledge, only two negative-sense RNA viruses have been rescued from negative-sense constructs, these are rabies and Sendai virus [17, 38]. The advantage of reverse genetic systems based on negative-sense constructs is that it excludes any possibility of MG cassette basal expression independent of the N, P, and L proteins.

The MG cassette together with the genes of the three helper proteins N, P, and L were independently cloned into pTRAk.2 vector and were agro-infiltrated into *N. glutinosa* and *N. benthamiana* using *A. tumefaciens*. Cloning the L gene in *E. coli* was initially problematic. Difficulty in cloning RNA virus genomes or viral genes in *E. coli* is common among the RNA viruses [24, 39–41]. The presence of prokaryotic promoters in viral sequences and spurious activity of eukaryotic promoters are found to be contributing factors for the observed instability [42, 43]. Strategies including the use of single point mutations, insertion of introns, and use of low-copy plasmid have been used to overcome these problems [26, 44, 45]. Nevertheless, success is predominantly a matter of trial and error as demonstrated by attempts to clone dengue virus type 1 (DV1) where a low-copy plasmid approach was suitable for cloning a DV1 replicon, but failed to produce the full-genome DV1 cDNA [46]. Ultimately, we succeeded in cloning the full-length L gene by incorporating three different introns into the 5' half of the gene and using an *E. coli* strain with modified *pcnB* gene for reduction of plasmid copy number [47]. While all three chosen introns have previously been demonstrated to be correctly spliced in their plant of origin [33–35], none have previously been tested as part of a cytorhabdovirus genome or in *N. glutinosa*. Insertion sites for introns were chosen between thymine and guanine nucleotides of ATG as described previously [25]. Introns were inserted at the 5' end of the gene to further enhance *in planta* mRNA translation [48]. Attempts to clone the full-length L gene using an intron-only approach (i.e., without plasmid copy number reduction), or solely by plasmid copy number reduction (i.e., without incorporation of introns) were unsuccessful (transformed *E. coli* either did not grow or yielded plasmids with large deleted sequences, although a number of strains were used with different growth conditions). A possible explanation is that disruption of cryptic sequences by introns and low template number (plasmid copy number) minimized inadvertent transcription [49]. Post-agroinfiltration, RT-PCR assessment performed on total RNA revealed the correct splicing of all three introns to yield mature mRNA, which was further confirmed by cDNA sequencing.

The first assessment undertaken with the assembled MG cassette was to confirm ribozyme auto-cleavage. The generation of exact 5' and 3' termini of vRNA enhances transcription and replication efficacy [50, 51]. Gel electrophoresis of MG RNA transcribed *in vitro* at 25°C suggested ribozyme auto-cleavage at both termini with the putative HDz auto-cleavage being more rapid than that of the HHz (which contradicts the results of previous studies [22]). Variation in ribozyme efficacy could have occurred due to differences in incubation temperature [52], and in future studies northern blotting with specific probes should be used to better identify the bands. From these *in vitro* results we expected ribozyme auto-cleavage and the production of MG precise ends to be replicated *in planta* at 25°C. Indeed, in previous studies, both ribozymes have demonstrated successful *in vivo* auto-cleavage in both cytoplasm and nuclei [53–55].

The first *in planta* assessment of the LNYV MG strategy was to investigate the transcription of the infiltrated MG *in planta* cassette (harbouring the DsRed ORF), N, P, and int-L genes. RT-PCR confirmed the presence of their respective mRNAs in samples co-infiltrated with MG, N, P, and int-L plasmids. Interestingly, int-L gene cDNA bands revealed the presence of mature and pre-mRNA. The presence of pre-mRNA may be due to intron-induced increase in transcription levels or an increase in mRNA stability [56, 57]. Further RT-PCR assessments and cDNA sequencing confirmed the correct splicing of all three introns. Any error in the intron splicing process would cause non-sense mediated mRNA decay or reading frame disruptions [58–60] and such mechanisms would have caused absence or loss-of-function of the product L protein. However, expression of the DsRed protein (discussed below) demonstrated the biological function of the L protein.

The success of the MG strategy was subsequently confirmed by detection of DsRed protein. Red fluorescence in leaf sections was observed using confocal microscopy (not shown), but significant background fluorescence, presumably from chlorophyll [61], prevented us from definitively confirming DsRed visualisation. In addition, as the MG cassette, N, P and int-L plasmids must all be in the same cell for DsRed to be produced, the concentration present may only have generated low levels of fluorescence. However, subsequent western blotting unequivocally demonstrated DsRed expression in agroinfiltrated *N. glutinosa* and *N. benthamiana*, generating bands of equivalent immunoreactivity to the positive control (Fig 4). These assessments revealed the presence of DsRed protein in samples co-infiltrated with four constructs encoding the MG cassette and the helper proteins but not in plant samples infiltrated with the MG cassette alone or with any the helper plasmids without the MG cassette. Independent expression of the DsRed protein from the MG construct should not be possible as the primary RNA transcripts are in the negative-sense. It is only through the biological interaction between the helper proteins and the *cis-* signals of the primary RNA transcript that the latter could be transcribed into positive-sense RNA, capped, polyadenylated and then translated into DsRed protein. The absence of DsRed protein in *N. glutinosa* or *N. benthamiana* infiltrated with the MG cassette alone provided further confirmation of the necessity for the biological interaction between LNYV *cis-* and *trans-* acting elements.

In conclusion, we have utilised introns to stabilise viral genomic sequence in *E. coli* and established a genomic-sense MG cassette system for a plant cytorhabdovirus, which will be a powerful tool for investigating virus replication and pathogenesis. Our work provides a platform for mutagenesis studies and functional analysis of the N, P, and L proteins and the *cis-* acting sequences of the virus genome. Furthermore, it is a successful proof-of-principle study that comprises the first step in the rescue of LNYV virus from cDNA.

## Supporting information

**S1 Table. List and description of primers used.** Primers were used for PCR amplification for *in vitro* RNA transcription, cloning, and RT-PCR screening.
(DOCX)

**S1 Raw images.**
(PDF)

## Acknowledgments

We are indebted to Dr Lorenzo Frigerio from the University of Warwick for his help with detection of DsRed protein. We are also grateful to Dr Thomas Rademacher and Dr Eva

Stoeger from Fraunhofer, IME who provided the transient expression vector with *Discosoma* sp. red fluorescent protein gene pTRAp-2G12-Ds.

## Author Contributions

**Conceptualization:** Ahmad E. C. Ibrahim, Craig J. van Dolleweerd.

**Funding acquisition:** Julian K-C. Ma.

**Investigation:** Ahmad E. C. Ibrahim.

**Methodology:** Ahmad E. C. Ibrahim, Craig J. van Dolleweerd.

**Project administration:** Ahmad E. C. Ibrahim, Craig J. van Dolleweerd, Pascal M. W. Drake, Julian K-C. Ma.

**Supervision:** Craig J. van Dolleweerd, Pascal M. W. Drake, Julian K-C. Ma.

**Validation:** Ahmad E. C. Ibrahim.

**Visualization:** Ahmad E. C. Ibrahim, Pascal M. W. Drake.

**Writing – original draft:** Ahmad E. C. Ibrahim, Pascal M. W. Drake.

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
