## [Decision Letter · Decision Letter 0]

4 Dec 2019

PONE-D-19-25109

Development of a minigenome cassette for Lettuce Necrotic Yellows Virus: a first step in rescuing a plant cytorhabdovirus

PLOS ONE

Dear Dr Drake,

Thank you for submitting your manuscript to PLOS ONE. After careful consideration, we feel that it has merit but does not fully meet PLOS ONE’s publication criteria as it currently stands. Therefore, we invite you to submit a revised version of the manuscript that addresses the points raised during the review process.

We would appreciate receiving your revised manuscript by Jan 18 2020 11:59PM. To enhance the reproducibility of your results, we recommend that if applicable you deposit your laboratory protocols in protocols.io, where a protocol can be assigned its own identifier (DOI) such that it can be cited independently in the future. For instructions see: http://journals.plos.org/plosone/s/submission-guidelines#loc-laboratory-protocols

We look forward to receiving your revised manuscript.

Kind regards,

Hanu R Pappu

Academic Editor

PLOS ONE

Journal Requirements:

1. PLOS ONE now requires that authors provide the original uncropped and unadjusted images underlying all blot or gel results reported in a submission’s figures or Supporting Information files. This policy and the journal’s other requirements for blot/gel reporting and figure preparation are described in detail at https://journals.plos.org/plosone/s/figures#loc-blot-and-gel-reporting-requirements and https://journals.plos.org/plosone/s/figures#loc-preparing-figures-from-image-files. When you submit your revised manuscript, please ensure that your figures adhere fully to these guidelines and provide the original underlying images for all blot or gel data reported in your submission. See the following link for instructions on providing the original image data: https://journals.plos.org/plosone/s/figures#loc-original-images-for-blots-and-gels.

Additional Editor Comments (if provided):

The MS was reviewed by three experts in the field and the majority opined that the research findings presented are worthy of publication in PLoS One, pending minor revisions. I agree with the majority recommendation and I encourage you to address the reviewers' comments and submit a revised version.

Reviewers' comments:

Reviewer's Responses to Questions

**Comments to the Author**

1. Is the manuscript technically sound, and do the data support the conclusions?

Reviewer #1: Yes

Reviewer #2: Yes

Reviewer #3: Yes

2. Has the statistical analysis been performed appropriately and rigorously? 

Reviewer #1: N/A

Reviewer #2: N/A

Reviewer #3: Yes

3. Have the authors made all data underlying the findings in their manuscript fully available?

Reviewer #1: Yes

Reviewer #2: Yes

Reviewer #3: Yes

4. Is the manuscript presented in an intelligible fashion and written in standard English?

Reviewer #1: No

Reviewer #2: Yes

Reviewer #3: Yes

5. Review Comments to the Author

Reviewer #1: The paper does not have any major scientific flaws but does not provide any interesting information either. The work presented has been done before alas not with LNYV. Is this good enough to warranty publication in PLoSOne? Not in my opinion. It just lacks originality.

Reviewer #2: The paper by Ibrahim and co-authors describes the set up of a replicating minigenome cassette based on genomic components of a cytorhabdovirus, lettuce necrotic yellows virus.

The paper is well written, with a well thought experimental plan, and the main conclusion seem to be well supported by the experiments carried out.

It needs to be improved a little bit, with further explanations that would make this paper useful for repeating experiments and future planning of the same work by other laboratories.

Here are my suggestions for improving it:

Line 19: maybe better “membrane-bound”?

Line 32: plasmids

Line 59-60: are you sure N. glutinosa is the choice plant species for studying virus biology and pathogenesis? The same can be said for N. benthamiana.

Line 140: N. glutinosa in Italics

In general, I think you need to specifically show in a figure (figure 2) a panel that shows the HHz-ribozyme sequence, with the initial viral sequence, and the putative cleaving site. This is important to show, and it would be better to show the secondary structure. No need to do so for the HDz)

Having said that, what you show in Fig.2.B is some evidence of cleaving (but not of the exact cleaving). Furthermore, cleaving is predicted based on RNA sizes (difficult to estimate in agarose gel, likely non-denaturing, even if not specifically stated in the Fig. 2 legend). So I would extensively use the word “putative” to describe the band assigned to the fragments predicted.

Line 174-175: it would be better to specifically mention the exact nt start and end of the 5’ and 3’ UTR referred to a GenBank reference viral sequence.

Line 251: specify better in which sense they were “unsuccessful”

Line 259-262: as I said above, you should have used northern blots with specific probes to identify better each of the three bands. I think this part of the discussion needs to be tuned down, or erased.

278-281: I am a little surprised about this. Probably working with appropriate filters and positive controls, you should have been able to distinguish dsRED from Autofluorescence. And even if I am not specifically asking for it, it would have been good to also have a GFP construct. Visual aspect of infection are also important, fur future steps when you will add putative movement proteins.

The arrows in Fig 1 are too big and misleading in some way. They need to be on the sides of the intron sites…. Not across the intron side.

Fig 4 should show some loading control (Coomassie or poinceu red).

Reviewer #3: Nice ms, describes difficult work that was well done, and of significant interest both to plant virologists and to plant molecular biotechnologists. I have very few concerns, and those are mainly use of taxonomic terms (see annotated ms). The English was excellent, the work was well described, and the results good enough to merit publication.

6. PLOS authors have the option to publish the peer review history of their article (what does this mean?). If published, this will include your full peer review and any attached files.

Reviewer #1: No

Reviewer #2: No

Reviewer #3: Yes: Edward P Rybicki

---

## [Author Response · Author response to Decision Letter 0]

7 Jan 2020

We thank the editor and the reviews for their comments and suggestions. Below are our answers to questions raised either in the text of the email sent on 04/12/2019 or the attached PDF. 

5. Review Comments to the Author

Reviewer #1: The paper does not have any major scientific flaws but does not provide any interesting information either. The work presented has been done before alas not with LNYV. Is this good enough to warranty publication in PLoSOne? Not in my opinion. It just lacks originality.

We respectfully disagree with this statement. In terms of manuscript originality, this is the first time a minigenome cassette strategy has been developed for LNYV. The Sonchus yellow net rhabdovirus (SNYV) publication (Wang et al. 2015) was based on a positive-sense construct, whilst our work in the present paper is based on negative-sense construct. In addition, SYNV is a nucleorhabdovirus whereas LNYV is a cytorhabdovirus. Finally, we believe that our strategy for solving viral gene instability in E. coli will be of considerable benefit to researchers.

Reviewer #2: The paper by Ibrahim and co-authors describes the set up of a replicating minigenome cassette based on genomic components of a cytorhabdovirus, lettuce necrotic yellows virus.

The paper is well written, with a well thought experimental plan, and the main conclusion seem to be well supported by the experiments carried out.

It needs to be improved a little bit, with further explanations that would make this paper useful for repeating experiments and future planning of the same work by other laboratories.

Thank you for the supportive comments. We address your comments below and in the manuscript. 

Here are my suggestions for improving it:

Line 19: maybe better “membrane-bound”? This has been changed as requested. 

Line 32: plasmids This has been changed as requested.

Line 59-60: are you sure N. glutinosa is the choice plant species for studying virus biology and pathogenesis? The same can be said for N. benthamiana. Yes, N. glutinosa is the plant species of choice for this virus. In addition to the publications cited in our manuscript, others include: (1) B. Harrison and N. Crowley, "Properties and structure of lettuce necrotic yellows virus," Virology, vol. 26, no. 2, pp. 297-310, 1965. or (2) T. Chambers and R. Francki, "Localization and recovery of lettuce necrotic yellows virus from xylem tissues of Nicotiana glutinosa," Virology, vol. 29, no. 4, pp. 673-676, 1966. or (3) B. Wolanski, R. Francki, and T. Chambers, "Structure of lettuce necrotic yellows virus: I. Electron microscopy of negatively stained preparations," Virology, vol. 33, no. 2, pp. 287-296, 1967.

We have included N.benthamiana as it is the species of choice for plant molecular farming using transient expression and believe that a positive result in the two species strengthens the paper.

Line 140: N. glutinosa in Italics. This has been changed as requested.

In general, I think you need to specifically show in a figure (figure 2) a panel that shows the HHz-ribozyme sequence, with the initial viral sequence, and the putative cleaving site. This is important to show, and it would be better to show the secondary structure. No need to do so for the HDz). We have added this as requested. We have also added “The MG cassette consisted of tr – multiple cloning site (MCS) – le cloned between hammerhead and hepatitis delta and under the control of T7 promoter (Fig. 2A). We also changed the first 8 nucleotides of HHz (accession number GU299211) to 5’ UCGUCCGU-3’, to allow base-pairing with the first 8 nucleotides of the LNYV tr (5’ ACGGACGA-3’) as indicated in a previous study (18)” to the manuscript.

Having said that, what you show in Fig.2.B is some evidence of cleaving (but not of the exact cleaving). Furthermore, cleaving is predicted based on RNA sizes (difficult to estimate in agarose gel, likely non-denaturing, even if not specifically stated in the Fig. 2 legend). So I would extensively use the word “putative” to describe the band assigned to the fragments predicted. We have added ‘putative’ into number of places in the manuscript including in the Fig2 legend. “(C) Agarose gel showing putative bands of tr – MCS - le fragment (358 nt), HHz - tr – MCS - le (412 nt), and HDz - T7T (233 nt).” 

Line 174-175: it would be better to specifically mention the exact nt start and end of the 5’ and 3’ UTR referred to a GenBank reference viral sequence. This has been changed as requested. “The coding region was cloned between LNYV L gene 3’ UTR (extending between 12,564 and 12,613nt of LNYV genome accession number AJ867584) and N gene 5’ UTRs (extending between 92 and 168nt).” 

Line 251: specify better in which sense they were “unsuccessful” This has been specified as requested. “transformed E. coli either did not grow or yielded plasmids with large deleted sequences, although a number of strains were used with different growth conditions)”

Line 259-262: as I said above, you should have used northern blots with specific probes to identify better each of the three bands. I think this part of the discussion needs to be tuned down, or erased. We have modified this line as follows: “Gel electrophoresis of MG RNA transcribed in vitro at 25°C suggested ribozyme auto-cleavage at both termini with the putative HDz auto-cleavage being more rapid than that of the HHz (which contradicts the results of previous studies (22)). Variation in ribozyme efficacy could have occurred due to differences in incubation temperature (52), and in future studies northern blotting with specific probes should be used better identify the bands.

278-281: I am a little surprised about this. Probably working with appropriate filters and positive controls, you should have been able to distinguish dsRED from Autofluorescence. And even if I am not specifically asking for it, it would have been good to also have a GFP construct. Visual aspect of infection are also important, fur future steps when you will add putative movement proteins.

We agree with this statement. DNA sequencing confirmed that the dsRED construct was correct - low expression levels in the plant may have been the reason for our difficulty in distinguishing between dsRed fluorescence and autofluorescence. Unfortunately, time and funding constraints prevented further exploration, however this will be a priority in future studies. We do believe that the western blotting results presented are conclusive. 

The arrows in Fig 1 are too big and misleading in some way. They need to be on the sides of the intron sites…. Not across the intron side. This has been changed as requested.

Fig 4 should show some loading control (Coomassie or poinceu red).

Unfortunately, we are do not have these data, however assessments were undertaken with leaf samples of the same mass treated under identical conditions. 

Reviewer #3: Nice ms, describes difficult work that was well done, and of significant interest both to plant virologists and to plant molecular biotechnologists. I have very few concerns, and those are mainly use of taxonomic terms (see annotated ms). The English was excellent, the work was well described, and the results good enough to merit publication.

Thank you for these supportive comments.

---

## [Editor Report · Decision Letter 1]

18 Feb 2020

Development of a minigenome cassette for Lettuce necrotic yellows virus: a first step in rescuing a plant cytorhabdovirus

PONE-D-19-25109R1

Dear Dr. Drake,

We are pleased to inform you that your manuscript has been judged scientifically suitable for publication and will be formally accepted for publication once it complies with all outstanding technical requirements.

With kind regards,

Hanu R Pappu

Academic Editor

PLOS ONE
---

## [Editor Report · Acceptance letter]

21 Feb 2020

PONE-D-19-25109R1 

Development of a minigenome cassette for Lettuce necrotic yellows virus: a first step in rescuing a plant cytorhabdovirus 

Dear Dr. Drake:

I am pleased to inform you that your manuscript has been deemed suitable for publication in PLOS ONE. Congratulations! Your manuscript is now with our production department. 

With kind regards,

on behalf of

Dr. Hanu R Pappu 

Academic Editor

PLOS ONE